# Benefits of Printed Graphene with Variable Resistance for Flexible and Ecological 5G Band Antennas

**DOI:** 10.3390/ma15207267

**Published:** 2022-10-18

**Authors:** Alexander G. Cherevko, Alexey S. Krygin, Artem I. Ivanov, Regina A. Soots, Irina V. Antonova

**Affiliations:** 1Department of Physics, Siberian State University of Telecommunications and Informatics, 86 Kirov Str., Novosibirsk 630102, Russia; 2Rzhanov Institute of Semiconductor Physics SB RAS, 13 Lavrentev Av., Novosibirsk 630090, Russia; 3Department of Semiconductor Devices and Microelectronics, Novosibirsk State Technical University, 20 K. Marx Av., Novosibirsk 630073, Russia

**Keywords:** graphene, antenna, electrodynamic characteristics, 2D printing, 5G standard

## Abstract

The possibility of creating antennas of the 5G standard (5.2–5.9 GHz) with specified electrodynamic characteristics by printing layers of variable thickness using a graphene suspension has been substantiated experimentally and by computer simulation. A graphene suspension for screen printing on photographic paper and other flexible substrates was prepared by means of exfoliation from graphite. The relation between the graphene layer thickness and its sheet resistance was studied with the aim of determining the required thickness of the antenna conductive layer. To create a two-sided dipole, a technology has been developed for the double-sided deposition of graphene layers on photographic paper. The electrodynamic characteristics of graphene and copper antennas of identical design are compared. The antenna design corresponds to the operating frequency of 2.4 GHz. It was found that the use of graphene as a conductive layer made it possible to suppress the fundamental (first) harmonic (2.45 GHz) and to observe radiation at the second harmonic (5.75 GHz). This effect is assumed to observe in the case when the thickness of graphene is lower than that of the skin depth. The result indicates the possibility of changing the antenna electrodynamic characteristics by adjusting the graphene layer thickness.

## 1. Introduction

The widespread introduction of the Internet of Things (IoT) poses the challenge of introducing new materials into telecommunications instead of such conductors as copper and silver. These materials are subject to the requirements of environmental friendliness and low cost, along with the requirements of high electric conductivity and mechanical flexibility. Nowadays, flexible and wearable electronics for individual medicinal devices, remote monitoring of different parameters, as well as for ensuring safety in emergency situations are intensively developed. According to estimates, graphene satisfies the basic requirements for antenna materials. Moreover, graphene conductivity has been continuously increased due to the progress in graphene fabrication technologies [1,2,3]. Graphene flakes can be used as the basic material for printing, which allows using a relatively cheap tunable 2D printing technology (inkjet and screen). So, these technologies are widely used for the creation of antennas [2,3,4,5,6].

In the decimeter range, the depth of the electric field penetration into films printed using graphene flakes is tens of microns. Therefore, the authors developing graphene antennas tried to make them thick enough so that they were similar in their parameters to metal ones and had low resistance, for example, [4,7] (where the printed graphene layer thickness was 50 μm or more). With this approach, the relatively high graphene resistivity served as an interfering factor that distorted the electrodynamic characteristics, making these antennas rigid enough. At present, the development and transition to flexible electronics in many areas of applications is a new and strategically important trend in modern progress [8,9,10,11]. Moreover, for terahertz wireless communications, thin (30 nm) graphene layers can be utilized in ultra-wideband micro-sized circular patch-shaped antennas [3,12]. An antenna based on CVD (Chemical vapor deposition)-grown graphene monolayer also has radiation in the terahertz range (2–3 THz) [13,14]. For the gigahertz range, which is in demand for intensively implemented IoT systems, 2D printed graphene technology has justified itself, including screen printing technology, where the antenna thickness varies from 6 to 35 microns. For example, the graphene-based printed 2.4 GHz dipole antenna was demonstrated in Refs. [15,16,17], where it is shown that these antennas are promising for IoT applications, and their characteristics are comparable to those of a similar copper antenna. A flexible and low-profile dual-band antenna based on highly conductive graphene-assembled films for 5G Wi-Fi applications and the operating frequency of 2.4–2.45 GHz and 5.15–7.1 GHz is proposed in Refs. [5,18,19]. Finally, the designs of graphene antennas allow one to vary operation in the frequency range of 2–14 GHz [1,19,20,21]. Moreover, it should be noted that the 5.7–5.8 GHz band, which is part of the 5G standard, is in demand of the Internet of Things both for smart cities and for the transmission of wearable electronics in order to transmit data from a person.

In the present study, the point of view on the relatively high graphene layer resistivity as a negative factor has been changed. The aim of this study is an experimental demonstration of the possibility of changing the antenna’s main electrodynamic parameters using increased graphene resistivity, as well as the possibility of controlling the antenna parameters and achieving new qualities. Increasing the layer resistance due to a decrease in their thickness in comparison with that of the skin depth is assumed to be the main factor in managing an antenna’s operating frequency. As a result, for the graphene antennas, the suppression of the first harmonic (2.45 GHz) and the observation of radiation at the second harmonic (5.75 GHz) was found.

## 2. Materials and Methods

The basis for the ink used for antennas was a suspension of the few-layer graphene flakes with a thickness of 0.5–3 nm and a lateral size less than 2 μm. The graphene suspension was created from natural purified graphite using electrochemical exfoliation and subsequent processing with an IKA Ultra-Turrax T18 digital laboratory disperser (IKA^®^-Werke GmbH & Co. KG, Staufen, Germany) in a water–ethanol solution at about 20,000 rpm. The methods of a graphene suspension fabrication are described in more detail in [22]. The graphene content of 2 mg/mL in a solution of 70% ethanol and 30% water was used. The resistance of the antenna active layers was measured by the four-point probe system using the JANDEL probe station and HM21 Test Unit (JANDEL Engineering Ltd., USA).

Printing electronics with liquid-phase exfoliated graphene flakes is a promising way for the cheap and high-performance fabrication of antennas. Two-sided symmetrical dipoles were fabricated by the screen printing from graphene flakes and from a copper layer. Details of the antenna design are given below in Figure 1. As a result, the identical graphene–graphite antennas with different thickness values of the active layer and copper, Cu, were fabricated for experimental studies. Antennas were marked as AGr and ACu according to the active layer used. For the ACu antenna, the conductive layer was cut out of foil and glued onto the cardboard substrate. Kodak photographic paper was chosen as the substrate for AGr. The choice of different substrates for AGr and ACu antennas is justified the same dielectric constant value the cardboard and photographic paper (2.31), which, in our opinion, guaranteed the same conditions for the electromagnetic wave propagation in the decimeter range. A preliminary comparison of the AGr antenna created on photographic paper and cardboard showed similar main parameters. For reasons of better adhesion and a weak substrate effect, different substrates for AGr and ACu antennas were used. The studies of antenna characteristics were carried out at the frequencies of the first and the second harmonics radiation (2.6 GHz and 5.7 GHz). The voltage standing wave ratio (VSWR), as one of the most demanded characteristics, was calculated and measured in the frequency range of (0.5–8) GHz.

The computer simulations for the graphene and copper antennas were carried out using the CST Studio Suite package (version of 2021). This package allows one to carry out high-precision 3D modeling of the design of objects for operation in the microwave range, as well as to study their electrodynamic characteristics in accordance with the design. The characteristics of both ideal and real objects (taking into account the energy losses) were modeled.

## 3. Results and Discussion

The 60 nm thick films made of graphene suspension on a SiO_2_/Si substrate had a resistance of ~50 Ω/sq, and the surface roughness on the area of 2 μm × 2 μm was 3 nm (see Figure 2a). Printing on photographic paper with strong surface relief results in significantly higher resistances and surface relief of the printed films. The sheet resistance of the graphene films depends on their thickness, as shown in Figure 3 (see also [22]). For the layer created by graphene screen printing, the resistance R_Gr_ demonstrates a strong non-linear dependence on layer thickness, with the R_Gr_ change in five orders of magnitude on the scale of 10^7^–10^2^ Ohm/sq. Measurements of resistance at different points of the antenna with a thickness ≥200 nm showed good reproducibility (±17%) along the antenna area. The antenna edge relief did not exceed 50 μm, as can be seen in Figure 2b.

The sheet resistance of the printed graphene layer, R_Gr_, in a wide range of layer thicknesses from 3 to 10^4^ nm, is presented on the double-logarithmic scale in Figure 3. For graphene this dependence is described by expression Lg(R_Gr_) = (4.758)/(Lg(d))^0.557^. The deviation of the curve for graphene from a linear dependence is attributed to the variation of the graphene layer resistivity with their thickness. Strong variation in the resistivity of a graphene–graphite layer on the thickness was theoretically justified in Ref. [23]. It is well-known that the Cu resistivity does not depend on the layer thickness and is equal to σ_Cu_ = 5.96 × 10^7^ S/m except for the near-surface skin depth with higher resistance and a thickness of several Angstroms. As a result, the volume part of Cu resistance R_Cu_ = f(d) in the double-logarithmic scale of Figure 2 is linear. For easy comparison, the sheet resistance of copper was multiplied by a factor of 10^5^. Thus, the graphene curve was obtained from experimental data; for copper, the curve was calculated. 

The strong graphene-graphite layer resistivity dependence on the thickness opens up new possibilities for the antenna design by means of changing the active layer thickness. This statement is demonstrated in the present study on the example of a two-sided symmetrical dipole in which, using the layer printed from graphene flakes, the first radiation harmonic was suppressed and significant radiation at the second harmonic (which, typically, should be negligibly low) was found. All modeling and experimental studies were carried out on graphene and copper antennas, which are of identical design.

### 3.1. Rationale for the Antenna Design Choice

To demonstrate the possibility of controlling the antenna electrodynamic characteristics using the graphene printing technology, the antenna design that could be used as an element of the Internet of Things in the 5G range of mobile operators was chosen. At present, the frequency band of 5.2–5.9 GHz is being mastered for these purposes. Graphene is an environmentally friendly material. That allows one to make the antenna environmentally friendly by using an environmentally friendly substrate. Environmental friendliness is one of the main requirements for IoT elements because, according to [24], the mass of electronic waste in the world by 2030 may exceed 70 million tons. The antenna substrate must be flexible, and that allows one to take advantage of graphene and the multigraphene layers, which demonstrate the stability of the layers with a bending radius of ~2–4 mm without losing their properties [25,26,27]. A symmetrical dipole was chosen as the antenna type. Firstly, it is well-studied, which makes it possible to control its electrodynamic parameters. Secondly, it is widely used in the 5G range [4,7]. The disadvantage of a symmetrical dipole is its relatively large dimensions, which can be reduced by making a two-sided dipole.

With the help of computer simulation in the CST package, the design of antennas, calculation, and study of their electrodynamic characteristics were carried out (see Figure 1). The conductive layer of the AGr antenna was screen printed. The active layer thickness was 7 μm, which was less than the skin layer thickness for operating frequencies. The Cu active layer thickness was significantly greater than the copper skin layer depth, as is customary for copper antennas. The printed dipole antenna contains a thin rectangular dielectric substrate of photographic paper (dielectric constant ε = 2.31, thickness 0.5 mm) with front and back surfaces (Figure 1b–e). On the front substrate surface, a slot line segment is made. Here, a solid rectangular printed conductor, which closes the slot line, is created. Moreover, on the front surface, on the opposite side of the slot line, dipole arms with a total length of 45.8 mm are made. The feeding microstrip is made on the reverse surface of the substrate as a J-shaped printed conductor. The antenna dimensions were optimized for a conductive metal layer in the CST high-frequency device numerical simulation package according to the criterion of minimum VSWR at a frequency of 2.6 GHz. The AGr antenna simulation was carried out while maintaining the dimensions and changing the properties of conductive printed surfaces. The thickness of the graphene conductive layer was d = 7 μm, while the graphene specific conductivity was set to σ = 4.4 × 10^4^ S/m.

### 3.2. Dependence of Powers on the Active Layer Conductivity

The total power supplied to the antenna (the standard term-incoming power) includes the power reflected from the input (reflected power) and the power received at the antenna (accepted power), which is spent on radiation (power radiated) and antenna losses (power losses) (Figure 3b). The manufactured ACu and AGr antennas are completely identical in their design. The difference lies in the power connectors, which do not make any noticeable difference in the measured characteristics. 

The antenna conductivity in the CST simulation package model was changed from 10 to 10^8^ S/m. The CST software package, using the power view calculation module, allows us to better understand the behavior of the antenna by analyzing the power balance. A power analysis was performed only for two frequencies, 2.6 and 5.7 GHz. A Gaussian stimulus waveform was used to perform power analysis, CST automatically calculates the appropriate excitation time pulse based on the frequency range setting. The CST transient solver works with time pulses that can be easily converted to the frequency domain using the fast Fourier transform.

The accepted and reflected powers’ dependence on the specific conductivity for ACu antennas is shown in Figure 4a,b. As it can be seen, for the specific conductivity of copper, the reflected power at a frequency of 2.6 GHz is 7 times less than that at a frequency of 5.7 GHz. At the same time, for the conductivity corresponding to the conductivity of AGr-antennas (σ~10^5^ S/m), there is good input matching for both frequencies. The reflected powers are not large, but the reflected power at a frequency of the 5.7 GHz is less than for the 2.6 GHz and is close to zero for the material with the conduction of graphene σ = (0.5–1.0) × 10^5^ S/m. In Figure 4c,d shows the distribution of the received antenna power, normalized to the power value. It can be seen (Figure 4c) that the difference between the radiation power and power losses for the two considered frequencies is insignificant.

### 3.3. Characteristics of Antennas

The characteristics of the antennas were measured with a complex of transmission and reflection vectormeters in an anechoic chamber. To eliminate the need for tuning when measuring the radiation pattern, RP, a P6-32 broadband measuring antenna, was used as a standard. To determine the gain factor, the comparison method with the reference measuring antenna was utilized. The data for the voltage standing wave ratio VSWR and the calculated and measured reflection coefficients S11 are shown in Figure 5.

As is seen in Figure 5a,b, for the AGr antenna, the minimum frequencies of the experimental and calculated VSWR are almost equal ($f_{minGr}^{exp}=5.75\;\mathrm{GHz}$ and $f_{minGr}^{calc}=5.7\;\mathrm{GHz}$ respectively), as well as the minimum VSWR values (1.06 and 1.18, respectively). The bandwidth determined at VSWR = 2 of the experimental AGr antenna is narrower than that of the simulated one. For a copper antenna, $f_{minCu}^{exp}=2.69\;\mathrm{GHz}$ and $f_{minCu}^{calc}=2.65\;\mathrm{GHz}$. However, the minimum values of the experimental and calculated VSWR are different (1.02 and 1.75, respectively).

The calculated reflection coefficient S11 for AGr and ACu is shown in Figure 5c,d. The S11 curves for Gr antennas confirm that the resonance at the frequency of 5.7 GHz is more pronounced, while for the ACu antenna, the resonance is at the frequency of 2.6 GHz. The resonance line is more pronounced in the case of S11. Computer simulations and experimental measurements of VSWR and S11 confirm the fact that the use of graphene as a conductive layer for the antenna will suppress the fundamental first harmonic of the antenna and increase the second harmonic intensity at the frequency of 5.7 GHz.

The normalized radiation patterns in polar coordinates in Figure 6 show that the correspondence between the experimental and calculated RPs is satisfactory, especially in the forward hemisphere, in the region of angles of 90 ± 60 degrees. Note that the main lobe of the RP of the experimental sample AGr antenna, in contrast to the main lobe ACu, antenna slightly deviates from the normal, and the contribution of the side lobes in the rear hemisphere increases, which does not significantly affect the performance of the AGr antenna at the frequency of 5.7 GHz. Small deviations of the RP AGr antenna most likely correspond to the influence of the power connector at a frequency of 5.7 GHz.

In the evaluation of the RP deviation of experimental samples from the RP of computer models, the given radiation patterns confirm that the operating frequency in the AGr antenna is the second harmonic of 5.7 GHz. The measurements also show that, within the passband, the RP is weakly dependent on the frequency. The appearance of directivity in the diagram of the AGr antenna at the frequency of 5.7 GHz is an unexpected experimental fact. A radiation pattern in the H-plane also confirms this statement. Experiments in the anechoic chamber with variable angle steps also confirm this result. The physics and the theory of this phenomenon have not been developed yet and require further studies.

### 3.4. Antenna Efficiency

The experimental results obtained for the radiation pattern and the gain factor of the AGr antenna made it possible to determine its efficiency for the 5.7 GHz band using the technique suggested in Ref. [28]. As the calculation showed, the efficiency of the AGr antenna is 11%, which indicates the operability of this antenna with the suppressed first harmonic in the 5.7 GHz band.

### 3.5. Comparison of Antennas with the Various Thicknesses and Resistances

Table 1 contains the values of skin depth calculated from the conductivity of the antenna layer. The list of cited works is far from complete because papers were selected from the table in the 5.7–5.8 GHz range, in which the authors reported the resistivity of the conductive antenna layer. The main part of the authors used an antenna design with a conductive layer thickness higher than the skin depth. Perhaps that is why the authors do not observe the effect found in the present work. Although, a specific antenna design is probably also needed to observe the effect. The exception is Ref. [29], in which the operation frequency was also 5.8 GHz and the thickness of the antenna layer was lower than the skin depth. It is worth mentioning that we are not finding a study which focuses on significantly changing the electrodynamic characteristics by selecting the thickness of the printed conductive layer and analyzing this promising aspect.

## 4. Conclusions

A prototype of the graphene flexible eco-friendly resistive antenna and its analogue with a copper conductive layer was created, and the electrodynamic characteristics of a graphene printed two-sided symmetric dipole and its copper analogue, created on the basis of computer simulation, were studied. It was shown that the variable resistivity of graphene leads to suppression of the first radiation harmonic of 2.6 GHz, which is the main one for a copper dipole. In this case, in the graphene dipole, the second harmonic of 5.7 GHz becomes the fundamental harmonic (band of cellular operators 5G). The result is confirmed by the experimental data for VSWR, reflection coefficients, and antenna pattern. The performance of the symmetric two-sided graphene dipole is confirmed by the experimental measurements and the calculation of the power distribution depending on the layer conductivity in the range of specific conductivity variation of eight orders of magnitude.

The dependence of the surface resistance of the printed graphene layer is non-linear. Its specific conductivity does not remain constant but depends on the layer thickness. Therefore, the result indicates the possibility of varying the antenna electrodynamic characteristics using the graphene layer with relatively high resistance and a thickness lower than the skin depth. The effect has been obtained using graphene with a relatively high resistivity, but the development of new graphene-based materials with lower resistivity could expand the capabilities of this method. Moreover, the use of thin graphene layers leads to the possibility of creating a flexible antenna.

## Figures and Tables

**Figure 1 materials-15-07267-f001:**
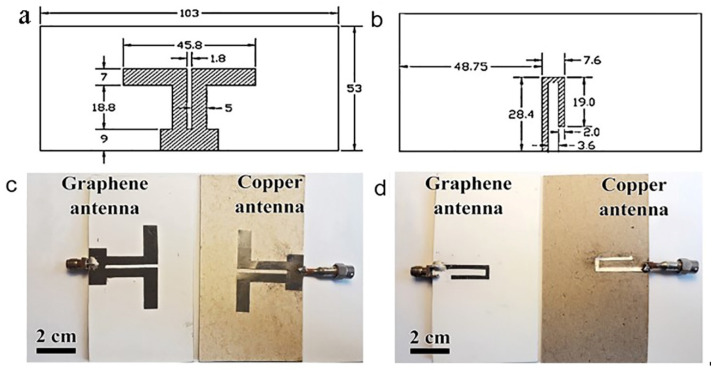
(**a**,**b**) Design of the front and reverse sides of antennas. The sizes are given in mm. (**c**,**d**) The photos of the front sides and the reverse (J-shaped power conductor) sides of ACu and AGr antennas.

**Figure 2 materials-15-07267-f002:**
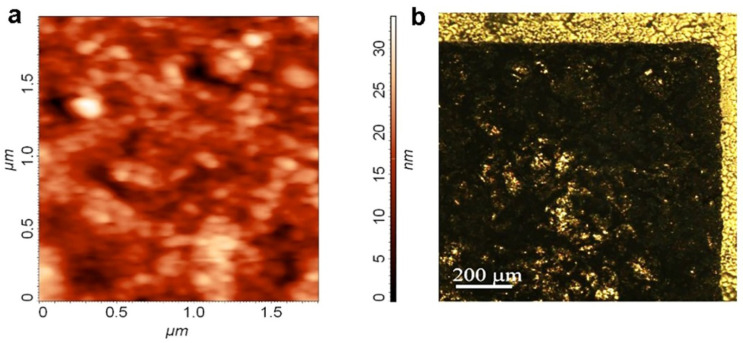
(**a**) An atomic force microscope image of a layer created by screen printing from a graphene suspension. (**b**) Optical image of the graphene suspension strip edge screen-printed onto Kodak photographic paper.

**Figure 3 materials-15-07267-f003:**
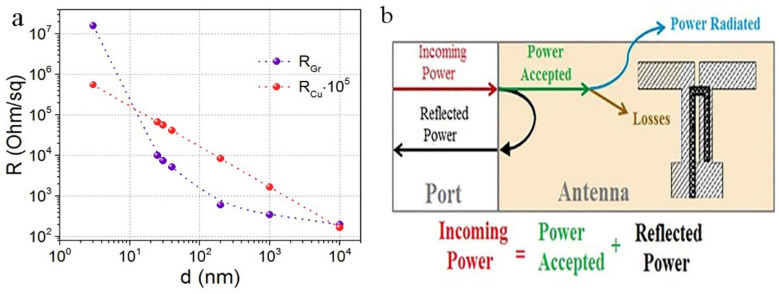
(**a**) Sheet resistances of the printed AGr layer and the resistance of Cu layer versus layer thickness. (**b**) Power distribution in the antenna.

**Figure 4 materials-15-07267-f004:**
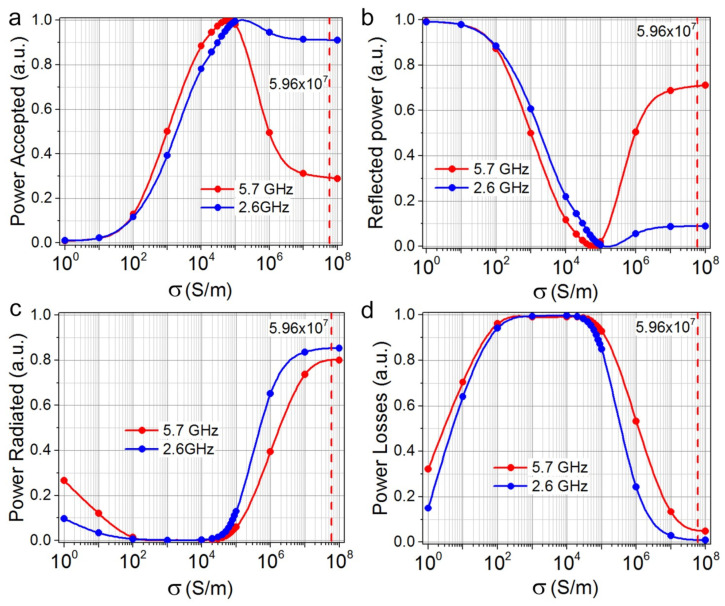
(**a**,**b**) Calculated dependence of the normalized powers in a two-sided symmetrical dipole of the ACu antennas versus the specific conductivity. The accepted and reflected powers are normalized to the incoming power value. (**c**,**d**) The calculated distribution of the power received by the AGr antenna depending on the specific conductivity. The powers are normalized to the accepted power value. The vertical line in the figures corresponds to the specific conductivity of copper: 5.96 × 10^7^ S/m.

**Figure 5 materials-15-07267-f005:**
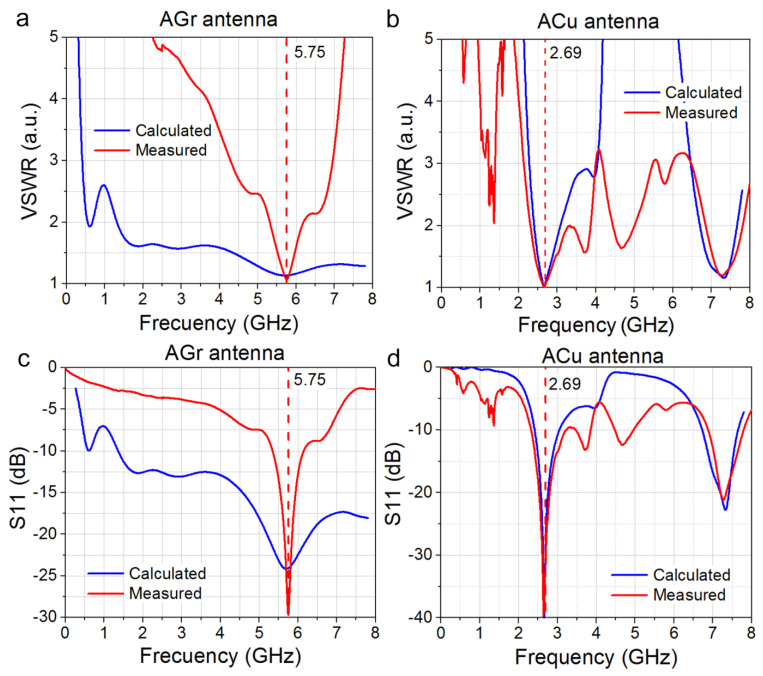
Frequency dependence of the (**a**,**b**) the voltage standing wave ratios VSWR; (**c**,**d**) the calculated and measured reflection coefficients S11 of AGr and ACu antennas.

**Figure 6 materials-15-07267-f006:**
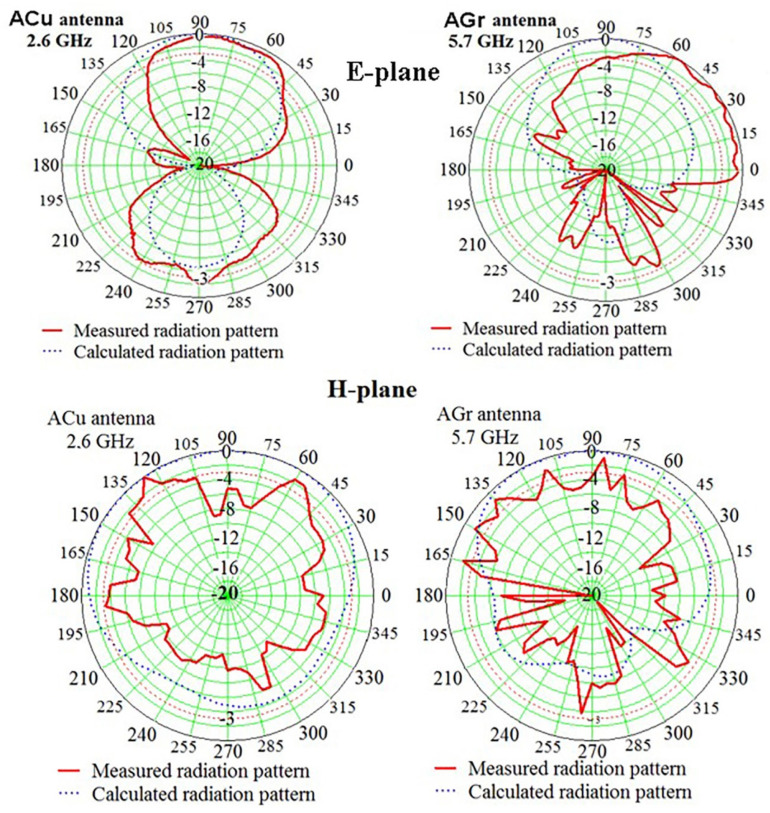
Radiation pattern of the ACu antenna for the frequency of 2.6 GHz and the AGr antenna for the frequency of 5.7 GHz in E- and H-planes. The frequency was chosen in accordance with the resonant peak position. The red dotted line corresponds to RP = −3 dB.

**Table 1 materials-15-07267-t001:** Comparison of antenna parameters with calculated from conductivity σ the value of skin depth.

Type of Antenna	Frequencies, GHz	Thickness, μm	σ, S/m	Skin Depth, μm	Ref.
Rectangular Patch Antenna	4–8	21	10^6^	5.6	[30]
Patch MIMO Antenna	3.5	21	1.1 × 10^6^	8.1	[31]
Patch Antenna	3.13–4.42	28	1.1 × 10^6^	7.2	[1]
Rectangular Patch Antenna	1.63	30	10^6^	12.5	[32]
Patch Antenna	5–13.5	–	4.1 × 10^4^	21.65	[19]
Patch Antenna	5.8	24	1.13 × 10^6^	6.2	[20]
Dual-Band CPW Graphene Antenna	2.4 5.8	240	3.5 × 10^5^	11.1	[5]
Dual-Band Conformal Antenna	2.4–2.45 5.15–7.1	24	1.13 × 10^6^	6.2	[17]
Patch Antenna	5.8	100	7.18 × 10^2^	246	[29]
Patch Antenna	5.8	24	1.13 × 10^6^	6.2	[20]
Dipole with Integrated Balun	5.75	7	4.4 × 10^4^	31.7	This study

## Data Availability

The data that supports the findings of this study are available from the corresponding author upon reasonable request.

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
