# Peer review of "Benefits of Printed Graphene with Variable Resistance for Flexible and Ecological 5G Band Antennas"

_materials, 2022, doi:10.3390/ma15207267_

Round 1

Reviewer 1 Report

General comment: In the manuscript “Benefits of the printed graphene variable resistance for flexible and ecological 5G band antennas”, the authors demonstrate the relationship between the thickness of the graphene layer and its sheet resistance, with the aim of determining the required thickness of the antenna conductive layer. The electrodynamic characteristics of graphene and copper antennas of identical design are compared. It was found that the use of graphene as a conductive layer made it possible to suppress the fundamental (first) harmonic and obtain radiation at the second harmonic. To create a two-sided dipole, a technology has been developed for the double-sided deposition of graphene layers on photographic paper. The result indicates the possibility of changing the antenna electrodynamic characteristics by adjusting the thickness of the graphene layer. The authors provided substantial experimental data and analysis, thus I recommend the manuscript could be considered pending a minor revision that addresses the following comments:

(1)  The authors declare that “Kodak photo-graphic paper was chosen as the substrate for AGr and a carton was used as a substrate for ACu antennas”. Here, the reasons why the substrate for AGr and ACu antennas is different ought to be presented.

(2)  In Figure 2The authors declare that “The curve for graphene was obtained from experimental data, for copper the curve was obtained by calculation”. Why the sheet resistances curve of the copper is gained by calculation not experimental testing, the authors should give the explanation.

(3)  In part 3.2, “The total power supplied to the antenna (the standard term incoming power) includes the power reflected from the input (reflected power) and the power received at the antenna (accepted power), which is spent on radiation (power radiated) and antenna losses (power losses) (Fig. 3b). Figure 3c, d shows the manufactured prototypes of graphene and copper antennas”. Here, whether the mark “Fig. 3b” is wrong, which may be “Fig. 3a”. similarly, the “Figure 3c, d” may ought to be “Figure 3d, e”, the authors should check and confirm.

(4)  In Figure 5, The authors declare that “The bandwidth of the experimental ACu is wider than that of the simulated one”. Why, what caused this result? The authors are supposed to give the detailed analysis.

(5)  Some articles on flexible materials, wearable electronics and IoT are informative for this work, which could be considered in the introduction to broaden the coverage, e.g., Adv. Funct. Mater., 2018, 28 (31), 1802989. Adv. Funct. Mater., 2018, 28 (47), 1805277. Adv. Energy Mater., 2021, 11 (31), 2101116. Adv. Energy Mater., 2020, 10 (36), 2001770.

Author Response

List of corrections

We have applied efforts to improve the manuscript according to reviewer reports and hope that in the present form everything becomes clear.  Thank you very much for the help with improving of our manuscript. English also had been corrected. We give all corrections in the text below with a yellow marking.

Reviewer 1

  • The authors declare that “Kodak photo-graphic paper was chosen as the substrate for AGr and a carton was used as a substrate for ACu antennas”. Here, the reasons why the substrate for AGr and ACu antennas is different ought to be presented.

Choice of the different substrates for AGr and ACu antennas is justified the same value of dielectric constant for the cardboard and photographic paper, which, in our opinion, guaranteed the same conditions for the propagation of an electromagnetic wave in the decimeter range. A preliminary comparison of the AGr created on photographic paper and cardboard showed the similar main parameters. For reasons of better adhesion and a weak substrate effect, the different substrates for AGr and ACu antennas were used. This explanation is added on page 2 in section 2 Materials and Methods.

  • In Figure 2,The authors declare that “The curve for graphene was obtained from experimental data, for copper the curve was obtained by calculation”. Why the sheet resistances curve of the copper is gained by calculation not experimental testing, the authors should give the explanation.

Experimental measurements of Cu layer is give value 35 Ohm/sq for thickness more than few Angstroms. So, it worth to compare RGr with the calculated volume part of RCu Now new variant of Fig. 2 description is given in page 3. 

(3)  In part 3.2, “The total power supplied to the antenna (the standard term incoming power) includes the power reflected from the input (reflected power) and the power received at the antenna (accepted power), which is spent on radiation (power radiated) and antenna losses (power losses) (Fig. 3b). Figure 3c, d shows the manufactured prototypes of graphene and copper antennas”. Here, whether the mark “Fig. 3b” is wrong, which may be “Fig. 3a”. similarly, the “Figure 3c, d” may ought to be “Figure 3d, e”, the authors should check and confirm.

      Thank you for the help. We have corrected all references on Fig. 3.

  • In Figure 5, The authors declare that “The bandwidth of the experimental ACu is wider than that of the simulated one”. Why, what caused this result? The authors are supposed to give the detailed analysis.

Broadening the bandwidth of the experimental antenna compared to that of the computer model is a typical situation, since the computer model is an idealized (simplified) antenna model. An idealized model without taking into account the influence of the input connector, the inaccuracies in dimensions, the spread of the permittivity of the substrate, etc.  These simplifications affect the simulation with a minimum step and time. In addition, when calculating in a wide frequency band, due to the peculiarities of the calculation algorithm, the CST package may inaccurately calculate VSWR values тear unity. To increase the accuracy, it is necessary to significantly increase the simulation time. Now we have optimized the model, taking into account these inaccuracies. As a result, this point of view was confirmed, the SWR and S11 ccurves for ACu, obtained taking into account the optimization result, are given in the Fig. 5b,d.

(5)  Some articles on flexible materials, wearable electronics and IoT are informative for this work, which could be considered in the introduction to broaden the coverage, e.g., Adv. Funct. Mater., 2018, 28, 1802989. Adv. Funct. Mater., 2018, 28 (47), 1805277. Adv. Energy Mater., 2021, 11 (31), 2101116. Adv. Energy Mater., 2020, 10, 2001770.

The authors are grateful to the referee for providing additional material. We have improved the Introduction and have added some Refs about flexible electronics.

Reviewer 2 Report

The authors present a paper titled Benefits of the printed graphene variable resistance for flexible and ecological 5G band antennas.

Regarding the present paper, I have the following comments:

1. The authors need enormously improve their English language and style.

2. The authors need to improve the introduction. Besides, they need to compare its work with other related works, for example:

Alharbi, A.G.; Sorathiya, V. Ultra-Wideband Graphene-Based Micro-Sized Circular Patch-Shaped Yagi-like MIMO Antenna for Terahertz Wireless Communication. Electronics 202211, 1305.

Hu, Z.; Xiao, Z.; Jiang, S.; Song, R.; He, D. A Dual-Band Conformal Antenna Based on Highly Conductive Graphene-Assembled Films for 5G WLAN Applications. Materials 202114, 5087

Morales-Centla, N.; Torrealba-Melendez, R.; Tamariz-Flores, E.I.; López-López, M.; Arriaga-Arriaga, C.A.; Munoz-Pacheco, J.M.; Gonzalez-Diaz, V.R. Dual-Band CPW Graphene Antenna for Smart Cities and IoT Applications. Sensors 202222, 5634. https://doi.org/10.3390/s22155634

3. It would be good if the authors explained how they obtained the charts shown in Figure 4. Or improve their explanation in section 3.2.

4. I agree that the VSWR is an essential parameter in antenna characterization, but a reflection coefficient chart could give more details about the resonance frequencies.

5. Why does the radiation pattern become directional at 5.7Ghz? What about the other radiation plain?

6. In line 78, include the name of the element, not just the chemical nomenclature.

7. In lines 70 and 150, you use a division sign to indicate the frequency range. It must be a hyphen or minus sing

8. In line 139, you wrote, "The supply microstrip line" It must be  "The feed microstrip."

 9. Please avoid employing sentences like in line 44  you wrote, "Of course."

Author Response

List of corrections

We have applied efforts to improve the manuscript according to reviewer reports and hope that in the present form everything becomes clear.  Thank you very much for the help with improving of our manuscript. English also had been corrected. We give all corrections in the text below with a yellow marking.

Reviewer 2

  1. The authors need enormously improve their English language and style

We have improved English language and style in the manuscript.

  1. The authors need to improve the introduction. Besides, they need to compare its work with other related works, for example:

Alharbi, A.G.; Sorathiya, V. Ultra-Wideband Graphene-Based Micro-Sized Circular Patch-Shaped Yagi-like MIMO Antenna for Terahertz Wireless Communication. Electronics 202211, 1305.

Hu, Z.; Xiao, Z.; Jiang, S.; Song, R.; He, D. A Dual-Band Conformal Antenna Based on Highly Conductive Graphene-Assembled Films for 5G WLAN Applications. Materials 202114, 5087

Morales-Centla, N.; Torrealba-Melendez, R.; Tamariz-Flores, E.I.; López-López, M.; Arriaga-Arriaga, C.A.; Munoz-Pacheco, J.M.; Gonzalez-Diaz, V.R. Dual-Band CPW Graphene Antenna for Smart Cities and IoT Applications. Sensors 202222, 5634. https://doi.org/10.3390/s22155634

We have improved the Introduction and have added the comparison with other antennas and references.

  1. It would be good if the authors explained how they obtained the charts shown in Figure 4. Or improve their explanation in section 3.2.

The antenna conductivity in the CST simulation package model was changed from 10 to 108 S/m. The CST software package, using the power view calculation module, allows us to better understand the behavior of the antenna by analyzing the overall power balance.  A power analysis was performed only two frequencies, 2.6 and 5.7 GHz. A Gaussian stimulus waveform was used to perform power analysis, CST automatically calculates the appropriate excitation time pulse based on the frequency range setting. The CST Transient Solver works with time pulses that can be easily converted to the frequency domain using the Fast Fourier Transform. The S-parameters can then be obtained from the resulting spectra in the frequency domain. The default Gaussian waveform guarantees a non-zero spectrum in the frequency domain band of interest, allowing accurate calculation of S-parameters.

T

  1. I agree that the VSWR is an essential parameter in antenna characterization, but a reflection coefficient chart could give more details about the resonance frequencies.

Considering that the VSWR and the reflection coefficient describe the same process, but focus on its different nuances, both characteristics are presented in the new edition of the manuscript (Fig.5).

  1. Why does the radiation pattern become directional at 5.7Ghz? What about the other radiation plain?

We have added in Fig.6 a radiation pattern in the H-plane. The appearance of directivity in the diagram of a graphene antenna at a frequency of 5.7 GHz is an unexpected experimental fact. Experiments in the anechoic chamber with variable angle step was confirm this result. The theory and physics of this phenomenon has not yet been developed and require the further studies.

  1. In line 78, include the name of the element, not just the chemical nomenclature.

We have added the name of the element in line 71 where Cu if mention in first time.

  1. In lines 70 and 150, you use a division sign to indicate the frequency range. It must be a hyphen or minus sing

It had corrected.

  1. In line 139, you wrote, "The supply microstrip line" It must be  "The feed microstrip."

It had corrected.

  1. Please avoid employing sentences like in line 44  you wrote, "Of course."

We have corrected this sentence and other places.

Round 2

Reviewer 1 Report

Thank you, this is a misspelling. In the References 10, the "Ren Z.i" should be "Ren Z".

Author Response

List of corrections

We have applied efforts to improve the manuscript according to reviewer reports. Thank you very much for the help with improving of our manuscript. English also was corrected. We give all corrections in the text with a yellow marking.

Reviewer 1

Thank you, this is a misspelling. In the References 10, the "Ren Z. i" should be "Ren Z".

We have corrected this misspelling and one more improve the English.

Reviewer 2 Report

There is still no exits comparison between the proposed work and other works.

The authors need to remark on the highlights of their research in the introduction.

Why in figure 5c the measured and calculated results are quite different? The authors need to explain this behavior.

Author Response

List of corrections

We have applied efforts to improve the manuscript according to reviewer reports. Thank you very much for the help with improving of our manuscript. English also was corrected. We give all corrections in the text with a yellow marking.

Reviewer 2

  1. There is still no exits comparison between the proposed work and other works.

We had introduced changes in the Introduction with the aim to demonstrate a shift in the antenna operation frequency caused by layer thickness. Of course, the design of antennas is also important for their functional parameters. Moreover, we had added the Table 1 for demonstration of the relation in layer thickness and skin-depth. This comparison gives an additional base for the main assumption that in the case when the thickness of graphene is lower than that of the skin-depth, the first radiation harmonic is suppressed, and radiation at the second harmonic (5.8 GHz) is observed.

  1. The authors need to remark on the highlights of their research in the introduction.

We had added highlights of ours research in the last paragraph of Introduction, corrected the Abstract and Conclusions.

  1. Why in figure 5c the measured and calculated results are quite different? The authors need to explain this behavior.

We have done additional efforts for optimization the calculation for graphene (Fig. 5c). Figure given below demonstrates that optimization by means of variation in the antenna geometry at 5% and variation in the conductivity up to 17%.

The figure and this answer are given in file “list of correction +paper” attached to answer.

Red line is the calculation line from the previous variant of Fig.5c

The blue dotted line is line after the optimization of the geometry and the conductivity.

Now a new variant of the calculated line is given in Fig.5c.

However, even after optimization, the discrepancy between experiment and calculation in a wide frequency range remains noticeable. From the point of view of the applicability of the antenna, the main role is played by the extremum region, where the discrepancies, from a practical point of view, are not important, because the resonance frequencies of the model and the experimental sample coincide, and the values of the minima of these curves differ by only 6 dB, which corresponds to the difference in the return power losses of 0.5%.

We think that the discrepancy in the curves is related to the lack of knowledge of the process of suppression of the first harmonic in a two-sided dipole antenna when using graphene with a thinner skin-depth. We intend to continue studying this effect by constructing equivalent antenna circuits and using graphene with different conductivities and thickness of the conductive layer.